# Binding Affinity of Some Endogenous and Synthetic TSPO Ligands Regarding the rs6971 Polymorphism

**DOI:** 10.3390/ijms20030563

**Published:** 2019-01-29

**Authors:** Neydher Berroterán-Infante, Monika Tadić, Marcus Hacker, Wolfgang Wadsak, Markus Mitterhauser

**Affiliations:** 1Division of Nuclear Medicine, Department of Biomedical Imaging and Image-guided Therapy, Medical University of Vienna, 1090 Vienna, Austria; neydher.berroteran@meduniwien.ac.at (N.B.-I.); monika_tadic@hotmail.com (M.T.); marcus.hacker@meduniwien.ac.at (M.H.); markus.mitterhauser@meduniwien.ac.at (M.M.); 2Department of Inorganic Chemistry, University of Vienna, 1090 Vienna, Austria; 3Center for Biomarker Research, CBmed, 8010 Graz, Austria; 4LBI Applied Diagnostics, 1090 Vienna, Austria

**Keywords:** TSPO, rs6971 polymorphism, diazepam binding inhibitor, protoporphyrin IX, disulfiram, binding affinity, radioligand

## Abstract

An intriguing target involved in several pathophysiological processes is the 18 kDa translocator protein (TSPO), of which exact functions remained elusive until now. A single nucleotide polymorphism in the *TSPO* gene influences the binding affinity of endogenous and synthetic TSPO ligands by facilitating a lower-affinity conformation, which modifies a potential ligand binding site, ultimately leading to a binding profile classification according to each genotype. For instance, some clinical effects of the distinctive binding affinity profile of cholesterol toward the TSPO of individuals with different genotypes have been extensively discussed. Therefore, we conducted an investigation based on a radioligand binding assay, to determine the inhibition constants of some reported endogenous TSPO ligands (diazepam binding inhibitor and protoporphyrin IX), as well as synthetic ligands (disulfiram and derivatives). We observed no dependency of the polymorphism on the binding affinity of the evaluated endogenous ligands, whereas a high dependency on the binding affinity of the tested synthetic ligands was evident.

## 1. Introduction

First described as a peripheral binding site of some benzodiazepines [1,2], the 18 kDa translocator protein (TSPO) has been identified as a key component of the outer mitochondrial membrane (OMM), where it contributes to important mitochondrial tasks. Its involvement in a multitude of physiological functions, such as synthesis of steroids and heme, cell differentiation, and apoptosis, has been recently explained on the basis of the modulation of cell nuclear gene expression via the mitochondria-to-nucleus signaling pathway. Furthermore, TSPO has been proposed to be dysregulated in a variety of diseases, including neurological and psychiatric disorders, several types of cancer and heart insufficiency [3,4,5,6,7,8]. Therefore, a plethora of endogenous and synthetic substances have been suggested as (more or less specific) TSPO ligands to fully characterize the pathophysiological role of this protein. For instance, cholesterol has been pointed out as the main endogenous TSPO ligand due to the involvement of TSPO in steroidogenesis. Other endogenous ligands include the tetrapyrroles, heme, and protoporphyrin IX (PPIX), as well as, the polypeptide diazepam binding inhibitor (DBI). Likewise, several benzodiazepines, isoquinolinecarboxamides, aryolxyanilides, and other classes of compounds are known as TSPO-targeting ligands [9,10,11]. Independently of their high or low binding affinity, TSPO ligands have demonstrated their ability to modulate, in different grades, several TSPO functions, and even have been proposed as therapeutic agents for TSPO-mediated diseases [12,13,14,15]. Interestingly, a single nucleotide polymorphism (SNP) in the *TSPO* gene (rs6971 polymorphism), which replaces alanine by threonine (Ala147Thr) in a non-conservative way, is known to influence the binding affinity of some TSPO ligands (endogenous and synthetic) according to the exhibited genotype. In detail, the binding affinities of such ligands vary among individuals with different genotype. Three affinity patterns related to this SNP can be distinguished as follows: High-affinity-binders (HAB, homozygotes), mixed-affinity-binders (MAB, heterozygotes), and low-affinity-binders (LAB, homozygotes) [16,17]. The nature of this distinctive binding affinity has been explained based on allosteric modulations observed in LAB [18]. Although this heterogeneous binding affinity has mainly affected the development of appropriate radioligands for positron emission tomography (PET), this polymorphism has also been associated to several anxiety-related clinical conditions [19,20,21,22]. The link between the rs6971 polymorphism and such disorders has been recently explained on the basis of the lower affinity of cholesterol toward the TSPO in LAB [23]. Since the rs6971 polymorphism plays a pivotal role in the molecular characterization of TSPO and consequently in the development of new TSPO ligands, we aimed to conduct a study to determine the influence of the rs6971 polymorphism on the binding affinity of two endogenous TSPO ligands: PPIX and DBI. In addition, disulfiram, a drug used for the treatment of alcoholism, and some of its derivatives were selected as synthetic TSPO ligands. This selection was based on a recent report by Yang et al., in which the disulfiram metabolite, dietyldithiocarbamate (DDC), was radiolabeled with the PET radionuclide copper-64 {[^64^Cu]Cu-DDC} and proposed as a TSPO PET ligand without any information regarding its sensitivity towards the rs6971 polymorphism [24].

## 2. Results

The endogenous ligands, protoporphyrin IX (PPIX), and diazepam binding inhibitor (DBI), as well as, the synthetic compounds disulfiram (DIS) and its derivatives: Sodium diethyldithiocarbamate (DDC), cupric diethyldithiocarbamate (Cu-DDC), methyl diethyldithiocarbamate (Me-DDC), *S*-methyl-*N*,*N*-diethyldithiocarbamate Sulfoxide (Me-DDC Sulfoxide), *S*-methyl-*N*,*N*-diethylthiocarbamate (Me-DTC), and *N*,*N*-diethyl-1-(methylsulfinyl)methanamide (Me-DTC Sulfoxide), were selected to perform radioligand binding assays by inhibiting the binding of [^3^H]PK11195 (a known TSPO ligand (*K*_d_ 29.25 nM), which binds to TSPO independent of the rs6971 polymorphism) in platelets membranes derived from subjects genotyped as HAB, MAB and LAB according to a well described assay [7]. Inhibition constants (*K*_i_) are shown in Table 1. All data fitted were to a single-site model.

## 3. Discussion

To the best of our knowledge, this is the first work reporting *K*_i_ values of the endogenous TSPO ligands PPIX and DBI, as a binding affinity indicator, within a setup using human material. All other studies involved the use of murine TSPO-expressing membranes. Nevertheless, our values were in agreement with studies, in which nanomolar and micromolar ranges were reported for the binding affinities of PPIX and DBI, respectively. Moreover, our results showed no dependency of the binding affinity of these two endogenous ligands regarding the rs6971 polymorphism. Hence, we propose that there will be no clinically relevant difference due to the biological binding of PPIX and DBI to TSPO of individuals of different genotype, as in the case of cholesterol [23]. 

Interestingly, affinities in the micromolar range were obtained for all disulfiram-related substances, including Cu-DDC. This is an interesting finding, due to the recent claim of [^64^Cu]Cu-DDC as a potential TSPO PET tracer, for which a very high affinity (nanomolar range) is normally required [25]. In this regard, we suggest a re-evaluation of this compound as a TSPO PET tracer, especially considering the proposed in vivo radiosynthesis (by injecting disulfiram and [^64^Cu]CuCl_2_ separately) [20], since the binding affinity of disulfiram is higher than the binding affinity of Cu-DDC, as shown in Table 1. Such evaluation should also consider the affinities of all potential disulfiram metabolites (DDC, Me-DDC, Me-DDC Sulfoxide, Me-DTC, and Me-DTC Sulfoxide) toward TSPO as they will compete for the molecular target in vivo. Remarkably, we found for this series, that only disulfiram itself exhibited a *K*_i_ value independent of the genotype group, whereas all derivatives displayed a strong dependency on the rs6971 polymorphism.

Although only few substances were tested, this study highlights the importance of evaluating the effects of the rs6971 polymorphism on the binding affinity of TSPO ligands. To this list of compounds, additional substances may be added. For instance, none of the benzodiazepines proposed as TSPO ligands have been thoroughly investigated to the best of our knowledge, regarding their binding affinities considering the rs6971 polymorphism so far.

In the future, it will be interesting to evaluate how these findings might influence the understanding of TSPO functionality in healthy and pathological tissues during the course of disease. It is clear that a thorough preclinical evaluation that also considers the specific binding affinities using human tissue samples is of great importance, particularly combining these findings with the residence time of ligands at the TSPO binding sites.

## 4. Materials and Methods

All chemicals were obtained from commercial sources (Merck KGaA, Darmstadt, Germany; TCI Deutschland GmbH, Eschborn, Germany; or Santa Cruz Biotechnology, Dallas, TX, USA). All solvents and chemicals were used as purchased without further purification. The inhibition constants of all substances were determined through competitive radioligand binding assays, using TSPO-expressing membranes obtained from previously genotyped subjects regarding the rs6971 polymorphism [7] and prepared as described elsewhere [16,17]. Briefly, venous blood (80 mL) was collected in EDTA-containing tubes from healthy volunteers identified as HAB, MAB, and LAB. Afterwards, whole blood was centrifuged at room temperature for 15 min (180× *g*). The resulting supernatant was centrifuged at room temperature for 15 min (1800× *g*) and a platelet pellet was obtained. Subsequently, the pellet was homogenized with buffer 1 (0.32 mM sucrose, 5 mM tris base, 1 mM MgCl_2_, pH 7.4, 4 °C) and centrifuged at 4 °C for 15 min (48,000× *g*) three times, with removal of the supernatant and re-homogenization of the pellet using buffer 2 (50 mM tris base, 1 mM MgCl_2_, pH 7.4, 4 °C; no sucrose) after each centrifugation step. Finally, the resulting membrane was mixed with 3 mL of buffer 2 and aliquots were stored at −80 °C until use. The total protein concentration of the membrane was determined via colorimetric detection of cuprous cation (Cu^1+^) by bicinchoninic acid (BCA) measuring the absorbance at 562 nm (Synergy HTX plate multimode reader, Biotek, Winooski, VT, USA), following the instructions manual of the BCA kit (ThermoFisher Scientific, Waltham, MA, USA).

Following the procedure described by Owen et al. [16], the radioligand binding assays were performed in test tubes, filled with 250 µL of the tested substance dissolved in assay buffer (50 mM tris base, 140 mM NaCl, 1.5 mM MgCl_2_, 5 mM KCl, 1.5 mM CaCl_2_, pH 7.4, 37 °C), 200 µL of the membrane suspension (250 µg.mL^−1^) and 50 µL of a 10 nM [^3^H]PK11195 (3149 TBq.mmol^−1^, Perkin Elmer, Waltham, MA, USA) solution. Non-specific binding was determined using 20 µM PK11195 instead of the tested substance, whereas total binding was assessed by incubating only [^3^H]PK11195 and membrane suspension. Incubations were carried out for 1 h at 37 °C. After this time, binding was terminated with ice cold wash buffer (50 mM tris base, 1.4 mM MgCl_2_, pH 7.4, 4 °C), while filtrating the membrane bound through GF/B glass fiber filters (Brandel, Gaithersburg, MD, USA) presoaked in assay buffer containing 0.05% polyethylene imine. In addition, 3 mL of wash buffer were passed through the filters and these were transferred into β-counting vials. Finally, 4 mL of β-scintillation cocktail (Perkin Elmer, Waltham, MA, USA) were added and the radioactivity was counted (Hidex 300SL beta counter, Hidex, Turku, Finland). Data from the competition plots were analyzed and IC_50_ and *K*_i_ values were calculated using a dissociation constant of 29.25 nM for [^3^H]PK11195 and GraphPad Prism^®^ software (GraphPad Software, San Diego, CA, USA).

## 5. Conclusions

In summary, this study highlights once again the influence of the rs6971 single nucleotide polymorphism on the binding affinity of endogenous and synthetic TSPO ligands. Although some advances have been made regarding the causes of the poor binding affinity of some ligands in LAB, there is no clear line of evidence in respect of the molecular requirements for homogenous binding affinity toward TSPO. Hence, in vitro evaluation of binding affinities seems necessary to achieve a comprehensive overview of the TSPO as well as for the development of new TSPO ligands.

## Figures and Tables

**Table 1 ijms-20-00563-t001:** Inhibition constants (*K*_i_) of the tested substances toward 18 kDa translocator protein (TSPO) using [^3^H]PK11195 as comparator (*K*_d_ 29.25 nM). Mean values ± SD are given (*n* = 3).

Substance	HAB (μM)	MAB (μM)	LAB (μM)
PPIX	0.033 ± 0.003	0.033 ± 0.001	0.035 ± 0.003
DBI	6.1 ± 0.9	6.0 ± 0.7	6 ± 1
DIS	2.2 ± 0.6	2.3 ± 0.2	2.6 ± 0.5
Cu-DDC	33 ± 2	47 ± 5	107 ± 6
DDC	136 ± 10	367 ± 9	531 ± 25
Me-DDC	166 ± 20	347 ± 30	574 ± 42
Me-DDC Sulfoxide	314 ± 5	1883 ± 186	2666 ± 253
Me-DTC	559 ± 13	2732 ± 80	3639 ± 324
Me-DTC Sulfoxide	2822 ± 206	4646 ± 763	7118 ± 972

HAB: high-affiniy-binders; MAB: mixed-affinity-binders; LAB: low-afinity-binders; PPIX: protoporphyrin IX; DBI: diazepam binding inhibitor; DIS: disulfiram; Cu-DDC: cupric diethyldithiocarbamate; DDC: sodium diethyldithiocarbamate; Me-DDC: methyl diethyldithiocarbamate; Me-DDC Sulfoxide: *S*-methyl-*N*,*N*-diethyldithiocarbamate Sulfoxide; Me-DTC: *S*-methyl-*N*,*N*-diethylthiocarbamate; Me-DTC Sulfoxide: *N*,*N*-diethyl-1-(methylsulfinyl)methanamide.

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
