# Peer review of "Binding Affinity of Some Endogenous and Synthetic TSPO Ligands Regarding the rs6971 Polymorphism"

_ijms, 2019, doi:10.3390/ijms20030563_

Reviewer 1 Report

In the introduction, as one of the functions of TSPO, please mention modulation of cell nuclear gene expression. e.g. Yasin et al., 2017 published by IJMS.

It is fascinating that low affinity TSPO ligands have differing affinities for HAB, MAB, and LAB, while high affinity TSPO ligands do not.  It would be interesting to discuss potential functional implications of this phenomenon.

Please also mention the functional significance of low affinity TSPO ligands. e.g. Vainshtein et al., 2015, published by Cell Death Discovery.

Costa et al., 2017, published by Chem Neurosci.

Chen et al., 2017, published by Stroke

I would like to suggest to the authors the consideration that the use of high affinity TSPO ligands for diagnositic purposes may very well interfere with the reported potential curative properties of low affinity TSPO ligands.

In general, it would be good to provide more thoughts regarding the potential implications of these findings for our understanding of TSPO functions.

Author Response

In the introduction, as one of the functions of TSPO, please mention modulation of cell nuclear gene expression. e.g. Yasin et al., 2017 published by IJMS.

Indeed, this is an interesting publication where the modulation of cell nuclear gene expression via the mitochondria-to-nucleus signaling pathway appears to explain the involvement of the numerous attributed functions to the TSPO. We added the reference as suggested by the reviewer and added the following text in the Introduction section:

Its involvement in a multitude of physiological functions, such as synthesis of steroids and heme, cell differentiation, and apoptosis, has been recently explained on the basis of the modulation of cell nuclear gene expression via the mitochondria-to-nucleus signaling pathway [3].

It is fascinating that low affinity TSPO ligands have differing affinities for HAB, MAB, and LAB, while high affinity TSPO ligands do not.  It would be interesting to discuss potential functional implications of this phenomenon.

This is in fact an important observation in our study, but it is not a general finding regarding the affinity of TSPO ligands. For instance, all second generation TSPO PET tracers do exhibit high affinity toward the TSPO (nanomolar range) and still high sensitivity toward the rs6971 single nucleotide polymorphism. Likewise, cholesterol also shows different (high) affinities regarding HAB and LAB. Hence, it would be too daring to draw generalized conclusions from the results of this limited study regarding potential functional implications for TSPO ligands.

Please also mention the functional significance of low affinity TSPO ligands. e.g. Vainshtein et al., 2015, published by Cell Death Discovery.

Costa et al., 2017, published by Chem Neurosci.

Chen et al., 2017, published by Stroke

Thanks for providing these references. They were added in the Introduction section together with the following section:

Independently of their high or low binding affinity, TSPO ligands have demonstrated their ability to modulate, in different grades, several TSPO functions and even have been proposed as therapeutic agents for TSPO-mediated diseases [12-15].

I would like to suggest to the authors the consideration that the use of high affinity TSPO ligands for diagnostic purposes may very well interfere with the reported potential curative properties of low affinity TSPO ligands.

Thank you for this interesting remark. For the application of diagnostic radiotracers in positron emission tomography the actual amount administered to the patients is in the pico- or even femto-molar range. Therefore, any pharmacological effect – both direct and indirect by interfering with therapeutic interventions – can be ruled out per se. So, there will be no blocking effect on available TSPO binding sites from a PET tracer regardless of its affinity, as TSPO has a high enough abundance.

In general, it would be good to provide more thoughts regarding the potential implications of these findings for our understanding of TSPO functions.

We thank the reviewer for this remark. To highlight the potential implications of the given findings, we added a final paragraph to the discussion section which reads:

In the future, it will be interesting to evaluate how these findings might influence the understanding of TSPO functionality in healthy and pathological tissues during the course of disease. It became obvious that a thorough preclinical evaluation also considering the specific binding affinities using human tissue samples is of great importance, particularly combining these findings with residence time of ligands at the TSPO binding sites.

Reviewer 2 Report

The Communication deals with the effect of rs6971 polymorphism in TSPO gene. The number of investigated persons was low, and ultimate conclusions can not be drawn. But the manuscript is of interest and could be published.

O suggest just some completion. Try to discuss the differences among genotypes in inhibition constants (Tab 1). You have mentioned it just in Introduction chapter (rr 51-53).

The differences HAB vs LAB are different (Cu-DDC 3.2 x; Me-DDc Sulfoxide 8.4 x, e.g.), do you suggest any explanation?

Author Response

I suggest just some completion. Try to discuss the differences among genotypes in inhibition constants (Tab 1). You have mentioned it just in Introduction chapter (rr 51-53).

To the best of our knowledge there is no generalized description of a correlation between prevalence of all TSPO-related diseases and the respective genotype of the patient population further to the citation given in the test already regarding anxiety disorders. It therefore would be purely speculative to make such a statement at this time-point. Furthermore, it was not within the scope of our work to evaluate clinical implications of binding profiles as such, although this would be fascinating to do in the future.

The differences HAB vs LAB are different (Cu-DDC 3.2 x; Me-DDc Sulfoxide 8.4 x, e.g.), do you suggest any explanation?

We are, at this moment, left without any reasonable explanation for this phenomenon. What is clear to us, is that there is a general trend (as expected) for lower affinities of some of the compounds for LAB in comparison to HAB. But the extent is obviously not a constant. One of the reasons might be that the biological material was taken freshly on each experimental day and then experiments for different compounds were also performed on different days. Therefore, a certain deviation might exist in the extent of the finding but not in the general conclusion. Further experiments in different labs using different donors should be conducted in the future to further substantiate these results and allow a general conclusion.